# GRILSS: Opening the Gateway to Global Reservoir Sedimentation Data Curation

Sanchit Minocha and Faisal Hossain

Department of Civil and Environmental Engineering, University of Washington, Seattle, WA-98105, USA
Corresponding Author: Faisal Hossain, fhossain@uw.edu

**Abstract.** Reservoir sedimentation poses a significant challenge to freshwater management, leading to declining storage capacity and inefficient reservoir operations for various purposes. However, trustworthy and independently verifiable information on declining storage capacity or sedimentation rates around the world is sparse and suffers from
inconsistent metadata and curation to allow global-scale archiving and analyses. The **G**lobal **R**eservoir **I**nventory of **L**ost **S**torage by **S**edimentation (GRILSS) dataset addresses this challenge by providing organized, well-curated and open-source data on sedimentation rates and capacity loss for 1,015 reservoirs in 75 major river basins across 54 countries. This publicly accessible dataset captures the complexities of reservoir sedimentation, influenced by regional factors such as climate, topography, and land use. By curating the information from numerous sources with disparate
formats in a homogenized data structure, GRILSS serves as an invaluable resource for water managers, policymakers, and researchers for improved sediment management strategies. The open-source nature of GRILLS promotes collaboration and contributions from the global community to grow the dataset. By providing essential reference data on sedimentation to understand the global challenge of reservoir sedimentation, this GRILLS dataset represents a gateway for the global community to share sedimentation and storage loss data for sustainable operation of world's
reservoirs for future generations.

## 1. Introduction

Artificial reservoirs created by damming a river or a lake are essential components of modern water management systems. Such reservoirs provide critical freshwater resources for irrigation, domestic and industrial use, as well as generate hydropower and moderate floods (McCully, 1996). However, these dams and their reservoirs pose significant
environmental challenges (Nguyen et al., 2024), including the destruction of ecosystems (Tundisi, 2018), loss of biodiversity (Wu et al., 2019), disruption of aquatic life (Morley, 2007), greenhouse gas emissions (Tran et al., 2023), and barriers to fish migration (Pelicice et al., 2015). Despite these drawbacks, numerous reservoirs remain in operation, as they can fulfill multiple essential functions for society. As dams age, the effectiveness of reservoirs becomes increasingly compromised by sedimentation, which gradually reduces their water storage capacity and threatens their
long-term functionality (Kondolf et al., 2014 and Wisser et al., 2013).

Hereafter, the terms dam and reservoir will be used interchangeably as shorthand for the dam-reservoir system. Despite the ongoing planning and construction of new dams worldwide (Zarfl et al., 2015), global reservoir capacity peaked in 2006 and has apparently been declining ever since (Wisser et al., 2013). This paradox is primarily due to the

accumulation of sediment at the bottom of reservoirs, which gradually reduces their storage capacity. Wisser et al. (2013) estimated that global per capita reservoir capacity declined by 19% between 1987 and 2010, and when sedimentation is factored in, the decrease is even more pronounced at 21%. Moreover, most of the optimal sites for dam construction have already been utilized, and the remaining feasible locations are often less economical (Morris & Fan, 1998). As a result, the new dams being built are generally smaller (Demirbas, 2009), and the rate of construction is not keeping pace with the rate of sedimentation. Overall, the current trend highlights the pressing need to address reservoir sedimentation to ensure a sustainable supply of freshwater for energy, food and flood control for future generations.

Regaining the storage capacity lost to sedimentation is essential for halting the global decline in reservoir capacity without having to build costly, uneconomical and ecosystem-damaging newer dams. One of the primary solutions is to implement sediment management techniques, such as sediment flushing, dredging, and watershed management, which requires a thorough understanding of sedimentation rates in reservoirs worldwide (Morris, 2014; Kondolf et al., 2014; George et al., 2017). Since sedimentation rates vary depending on regional factors, conducting localized studies is crucial for accurately assessing these rates. Topography, geology, vegetation and precipitation can affect the rate of soil erosion which can further affect sedimentation rates and consequently the rate of storage capacity loss. Due to the interplay of vast number of other factors such as operation procedure, reservoir shape and purpose, sedimentation rate can also be significantly different from one neighboring reservoir to another.

Sedimentation rates in reservoirs can be estimated using either direct or indirect methods (Nyikadzino & Gwate, 2021). Direct methods, such as bathymetric surveys, involve measuring the reservoir's depth over time and comparing the surveys to estimate storage loss, which is attributed to sediment deposition. These methods tend to be most accurate but are also more costly (Samaila–Ija et al., 2014; Patil et al., 2015). On the other hand, indirect methods use satellite remote sensing or numerical modeling to estimate sedimentation rates (Elçi et al., 2009; Hanmaiahgari et al., 2018; Pandey et al., 2016; Zhang et al., 2006). While indirect methods are less expensive, they rely on the availability of in-situ data, such as soil moisture and soil type for numerical models, or reservoir water elevation data, often obtained through in-situ or altimetry-based measurements, for remote sensing applications (Ghosh et al., 2024; de Oliveira Fagundes et al., 2020; Dutta, 2016).

Due to these limitations, only a small fraction of the world's millions of reservoirs have been thoroughly assessed or gauged for sedimentation rates, and even fewer are well-documented in published literature. Countries like the United States and India have made efforts to compile sedimentation databases for a limited number of reservoirs within their national boundaries (Gray et al., 2010; Central Water Commission, 2020). However, despite these initiatives, there remains no comprehensive, well-documented global dataset on reservoir sedimentation rates or storage capacity loss. While numerous regional studies have analyzed sedimentation rates for individual reservoirs, a unified, well-curated and publicly available global dataset capturing sedimentation rates and available storage capacity across the world remains absent. This lack of a cohesive data archive for the global community presents a major challenge in addressing

and understanding reservoir sedimentation on a global scale (Syvitski et al., 2003; Vanmaercke et al., 2011; Wisser et al., 2013; Perera et al., 2023).

In this paper, we address the need for a comprehensive, consistently curated and publicly available dataset that compiles the loss of storage capacity due to sedimentation in reservoirs worldwide. We introduce **G**lobal **R**eservoir **I**nventory of **L**ost **S**torage by **S**edimentation (GRILSS; Minocha & Hossain, 2024) dataset, the first open-source dataset of its kind, to the best of our knowledge. GRILSS includes information on over 1,000 reservoirs, detailing the amount of sedimentation and lost capacity over specific time intervals along with various meta data details.

Additionally, GRILSS provides other attributes such as the height of impoundment, catchment area, and original built capacity of the reservoirs, where available.

GRILSS has the potential to be highly valuable for the global freshwater community, offering insights into sedimentation patterns in reservoirs around the world. By providing sedimentation rates and their variations across

different river systems, GRILSS can potentially enhance our understanding of how sedimentation processes differ regionally. GRILSS can inform sediment management strategies by identifying how these techniques should be tailored to specific regions. As the first version of its kind, we anticipate that GRILSS will continue to grow over time, both in size and use, with future contributions on sedimentation data from the global community to expand the global coverage of reservoirs.

**2. Dataset Preparation**

**2.1 Collection of sedimentation data**

The GRILSS dataset (Minocha & Hossain, 2024) was compiled through an extensive literature review of published articles, theses, government documents, and websites, some of which were in languages other than English (Estrada et al., 2015). The literature review was not completely systematic. Search engines such as Google, Google Scholar,

and ResearchGate were utilized with keywords like "reservoir capacity loss," "reservoir sedimentation," "bathymetric surveys," and "sedimentation rate," among others. Conducting a fully systematic review proved challenging due to the inclusion of grey literature, which is often not indexed in standard databases or publicly accessible. After carefully reviewing hundreds of papers and documents, the dataset was created by manually compiling data from 143 different sources using Microsoft Excel. Only sources containing specific sedimentation data related to reservoirs were

included. This thorough process ensured that the dataset covers sedimentation information from a wide range of regions.

Since sedimentation can be reported in various ways, we tracked metrics from different sources. Many sources directly provided sedimentation amounts in million cubic meters (MCM) along with the duration over which the sedimentation

occurred. However, some sources reported sedimentation in million tons (MT) (Kim et al., 2014; Palinkas & Russ et al., 2019). Additionally, certain sources mentioned the percentage of capacity loss over a specific period but did not

always include the reservoir's original capacity (Güvel, 2021; Patro et al., 2022). A few sources only provided the sedimentation rate (either in MCM or MT), with or without specifying the time frame used to calculate it (Lorsirirat, 2014; Yang et al., 2022). Due to this variability in reporting, the GRILSS dataset had to be carefully compiled and standardized to ensure consistency of data structure across the dataset (Minocha & Hossain, 2024).

For sources where the sedimentation amount was reported in million tons (MT), the bulk sediment density was obtained either from the same source or a comparable source for that region. Using this bulk density, the sedimentation volume was then estimated in million cubic meters (MCM) with the help of Eq. (1). In cases where only the sedimentation rate was available, the rate was assumed to represent the sedimentation amount over a one-year period using Eq. (2). Unless otherwise specified, this duration was assumed representative from two years before the source's publication date to one year afterwards. For other reservoirs, the sedimentation rate was also estimated using Eq. (2).

$$Sedimentation\ volume\ (MCM) = \frac{Sediment\ weight\ (MT)}{Bulk\ sediment\ density\ \left(\frac{ton}{m^3}\right)} \tag{1}$$

$$Annual\ sedimentation\ rate\ \left(\frac{MCM}{year}\right) = \frac{Sedimentation\ volume\ (MCM)}{Time\ Duration\ (year)} \tag{2}$$

For cases where sedimentation was reported as a percentage of capacity loss, but the reservoir's original capacity was not provided, the initial capacity was extracted from other datasets or references. The sedimentation volume was then calculated using Eq. (3). Equation (3) was also used to calculate the percentage of capacity loss when sedimentation volume and original built capacity were available. The annual rate of reservoir capacity loss, expressed as a percentage of the original built capacity was estimated for all reservoir using Eq. (4). This approach ensured consistency in estimating sedimentation volumes, annual sedimentation rates, and capacity loss percentages across various reporting formats. However, sedimentation weight in million tons (MT) was not estimated for all reservoirs due to insufficient bulk sediment density data.

$$Sedimentation\ volume\ (MCM) = \frac{\%\ Capacity\ loss * Original\ built\ capacity (MCM)}{100} \tag{3}$$

$$Annual\ capacity\ loss\ rate\ \left(\frac{\%}{year}\right) = \frac{\%\ Capacity\ loss}{Time\ Duration\ (year)} \tag{4}$$

In the context of sedimentation volume, it should be noted that this is typically derived by comparing reservoir bathymetry data collected at two different times. The capacity lost is attributed to the sedimentation caused by sediment transport from the catchment and deposition within the reservoir. While this approach assumes sedimentation volume is primarily from catchment-derived input, it does not account for sedimentation resulting from the transport of sediments within the reservoir itself, including processes like the retreat of reservoir banks. Future studies addressing sedimentation dynamics should explicitly consider such cases when relevant.

140 While efforts were made to standardize sedimentation data through conversions and assumptions, such as assuming a one-year sedimentation period when not specified or estimating bulk sediment density, these approaches inevitably introduce uncertainties. It is important to note that the assumption of a one-year sedimentation period affects essentially sedimentation volume calculations and not sedimentation rates. Thus, sedimentation rates should be used for inter-reservoir comparisons. The reliance on such assumptions reflects the limitations of the available data and

145 underscores the need for more robust and consistent reporting practices globally. Consequently, trends and comparisons derived from these data should be interpreted with caution.

For consistency and accuracy in reporting reservoir sedimentation data, we recommend the inclusion of minimum key metrics for each reservoir. These should include the sedimentation volume in million cubic meters (MCM) and the

150 time frame over which sedimentation occurred. Additionally, it is important to indicate whether the reservoir employs any sediment management techniques. Key dam attributes such as the original built capacity, construction year, and precise dam coordinates should also be reported. Although variations in the spelling of reservoir names may arise across different sources, the inclusion of coordinates offers a unique and reliable method for identifying and comparing reservoirs, thus improving data consistency and reliability in future studies.

155 **2.2 Collection of reservoir attributes**

From each source, the reservoir name and the country of location were extracted and manually compiled in Microsoft Excel along with the sedimentation data. Other reservoir attributes, such as impoundment height, year of construction, original capacity, and catchment area, were also recorded when available.

160 While geospatial coordinates of the reservoir were rarely provided from most sources, they were included in the GRILSS dataset whenever available. In many cases, sources provided only a figure showing the approximate location of the reservoir, making it difficult to derive precise geospatial coordinates. Additionally, some sources provided only the extent of the reservoir rather than the exact coordinates of the dam impoundment. Out of the over 1,000 reservoirs for which data was collected, only around 170 had geospatial coordinates of the dam.


Since a reservoir database without georeferencing is of limited utility for analyses, addressing the challenge of mapping the reservoir attributes to their corresponding dam locations became crucial. To accomplish this, the Global Dam Tracker (GDAT), which contains more than 35,000 georeferenced dams with attributes, was utilized (Zhang & Gu, 2023). The reservoir names in GRILSS were fuzzy-matched to the dam names in GDAT using the Python library

170 "thefuzz" (Seatgeek, 2021). This method was used to account for minor variations in the spelling of reservoir names across different regions, where slight differences in spelling could prevent a direct match.

For each reservoir in the GRILSS dataset, the top 10 dam name matches, based on fuzzy-matching scores, were extracted from the GDAT dataset. Since there could be multiple dams or reservoirs with the same name, a manual

175 review of the matches was conducted. This involved comparing other attributes such as year of construction, original

capacity, and country for both datasets to ensure consistency. If these attributes aligned, the match was confirmed. Through this process, we successfully identified the corresponding dams for 613 GRILSS reservoirs from the GDAT dataset.

In cases where attributes like impoundment height, year of construction, or original built capacity were missing in GRILSS but available in GDAT, these values were added to the dataset. However, geospatial coordinates were not always available, even for dams listed in GDAT. For all remaining reservoirs lacking dam coordinates, geospatial data was manually derived using Google Earth and Google Maps, by searching for the reservoir name in the specified region as indicated by the original source. When approximate location maps were available, these figures were

overlaid on actual maps with proper scaling to accurately determine the impoundment locations. Through this process, all reservoirs in the GRILSS dataset were successfully georeferenced (Minocha & Hossain, 2024).

**2.3 Collection of other attributes**

To increase the utility of GRILSS, additional attributes were added to the dataset. One key attribute added from each source was the method used to estimate sedimentation for a given reservoir, categorized into three main types: (1)

bathymetric survey, (2) remote sensing, or (3) numerical modeling.

1    Bathymetric survey (BS): This method involves repeating a survey after a specific time period and comparing it with previous surveys. The observed loss in reservoir capacity is attributed to sedimentation volume (Stauch et al., 2024; Ugwu et al., 2021).

2    Remote sensing (RS): In this method, water elevation data is collected through altimetry or in-situ measurements, while the surface area of the reservoir is determined using optical satellite imagery. By combining these, an area-elevation curve of the reservoir is generated, which helps calculate the current capacity. The difference between this capacity and the original built capacity indicates the sedimentation volume (Pandey et al., 2016; Zhang et al., 2006).

3    Numerical modeling (NM): Sedimentation estimates can also be derived through hydrological and sediment models. These models use in-situ data such as inflow, sediment concentration, and sediment density to simulate sediment transport and deposition in the basin, providing an estimate of sedimentation (Elçi et al., 2009; Hanmaiahgari et al., 2018).

In terms of relative accuracy, the methods for estimating sedimentation rates and reservoir storage vary depending on the approach and data used. Bathymetric surveys are generally considered to provide the most accurate estimates, as they directly measure the reservoir's depth and volume. Satellite remote sensing accuracy is influenced by factors such as image resolution, the method used to estimate the surface area of the reservoir, and whether elevation data is in situ or derived from altimetry. Numerical modeling, on the other hand, depends heavily on the accuracy of input data and

the assumptions inherent in the model. While these methods have inherent limitations, bathymetric surveys typically

yield the highest accuracy, followed by remote sensing and numerical modeling, respectively (Nyikadzino & Gwate, 2021; Gao, 2009).

Another attribute incorporated into the GRILSS dataset is the classification of reservoir storage type, indicating
whether the sedimentation volume was estimated for live storage or gross storage for each reservoir. Additionally, Google Maps was utilized to verify whether the reservoir had been removed and to determine if it was associated with a creek dam or a dry dam. The website location (i.e., Uniform Resource Locator URL) for each reservoir's original data source was also documented in the dataset. For some reservoirs located in Zambia, the built year was roughly estimated based on sedimentation and capacity loss rates, with these assumptions noted in the comment section.

To ensure compatibility of the GRILSS dataset with one of the most widely used dam datasets, the Global Reservoir and Dam database (GRanD), a comparison was conducted to identify common reservoirs in both datasets (Lehner et al., 2011). This involved creating a 100-meter buffer around the dam locations in the GRILSS dataset. Dams within this buffered radius were then matched to those in the GRanD database.

As a result of the above data curation practices, the GRILSS dataset can now be effectively utilized alongside both the GDAT and GRanD datasets, with increased utility for research and analysis related to reservoir sedimentation, operations and management.

**2.4 Reservoir polygons**

Reservoir sedimentation varies significantly from one reservoir to another, with one key factor being the shape of the reservoir. Generally, sedimentation is higher when the reservoir's shape complexity—defined as the ratio of shoreline length (or wetted perimeter) to surface area—is greater. Conversely, reservoirs with a larger surface area but a lower wetted perimeter, indicating low shape complexity, tend to experience less sedimentation (Kantoush & Schleiss, 2014). Given the importance of reservoir geometry in sedimentation studies, we undertook efforts to extract the shape
polygons for each reservoir in the GRILSS dataset. These shape polygons have the potential to provide valuable insights for future analyses by the global water community.

OpenStreetMap (OSM) was used to extract reservoir geometry shapes for each reservoir based on the dam locations (OpenStreetMap, 2017). The Overpass API was employed to fetch shapes within a 1000-meter radius of each dam.
To ensure that only reservoir shapes were selected, the results were filtered based on specific tags such as "natural" = "water," "water" = "reservoir," or "water" = "lake" (Overpass, 2024). When multiple shapes were retrieved, the shape closest to the dam's point location was selected.

When applying the initial strategy, no reservoir shapes were found for around 300 reservoirs. Upon closer inspection
in QGIS (QGIS, 2024), with OSM as the basemap, it was discovered that for 200 of these reservoirs, the dam coordinates—mostly sourced from GDAT—were significantly misplaced. After manually correcting these

coordinates, reservoir shapes were re-extracted from OSM. However, for around 150 reservoirs, the shape remained hard to retrieve.

A quality check was then performed by reviewing each reservoir shape in QGIS using OSM and Google satellite imagery. It was found that for 150 reservoirs, no reservoir polygons existed in OSM, while for 20 reservoirs, the shapes were incomplete, and for 35 reservoirs, the wrong shapes were selected. Each of these cases was manually handled for quality control. For reservoirs with incorrect shapes, the correct OSM IDs were identified using the OSM webapp, and the shapes were re-fetched. For incomplete shapes, multiple OSM shapes were joined to create a single accurate geometry.

For reservoirs without any polygons in OSM, HydroLAKES was used to match dam locations to lake geometries, providing shapes for about 50 reservoirs (Messager et al., 2016). For the remaining reservoirs, the Global Surface Water Occurrence dataset (Pekel et al., 2016) was used, but the resulting shapes, derived from raster data, were pixelated. These shapes were manually refined to improve the overall quality of the GRILSS dataset. It is important to note that no reservoir geometry could be derived for dry dams, creek dams, or dams that had already been removed.

## 2.5 Catchment polygons of reservoirs

A major factor that directly impacts reservoir sedimentation is the catchment of the reservoir. Key catchment characteristics such as area, slope, land use, and land cover play a significant role in determining sedimentation rates. To enable users to assess these characteristics, we focused on extracting the catchment polygons for all reservoirs, as these shapes provide valuable insights for sedimentation analysis.

The Watershed (Spatial Analyst) tool in ArcGIS Pro was used to delineate catchments, utilizing a 90 m digital elevation model. To automate this process for over 1000 reservoirs, a geoprocessing workflow was created using Model Builder, a visual programming language. Snap distances were adjusted based on the original built capacity of the reservoirs, with larger reservoirs using larger snap distances (e.g., 1200 meters for those over 1 MCM and 50 meters for those under 0.1 MCM).

However, for a few reservoirs, particularly those in Taiwan, this method was unsuccessful. In those cases, catchment polygons were created using the Python library "pysheds" (Bartos, 2020) and HydroSHEDS flow direction data at 3 arc-seconds resolution (Lehner et al., 2008). Through this combined approach, catchment geometries were successfully obtained for all the reservoirs in the GRILSS dataset, which were then used to calculate catchment area when it was not available from the original source. The catchment shapefiles are provided as supplementary information for users. However, in cases where high-resolution DEMs are required, users are encouraged to use their own catchment geometries to suit the specific needs of their work.

## 3. Dataset Description

The Global Reservoir Inventory of Lost Storage by Sedimentation (GRILSS) dataset contains sedimentation data for over 1000 reservoirs worldwide. Compiling this dataset took sustained effort and methodical work over a year, involving detailed manual corrections, quality assurance, and quality control to ensure its accuracy. To the best of our knowledge, GRILSS is the first dataset of its kind, aimed at supporting global studies on reservoir sedimentation and its impact on water storage. The dataset is composed of four files. The main file, an Excel spreadsheet, contains historical records of reservoir sedimentation, with each record assigned a unique Observation ID (OID). A single reservoir can have multiple records, reflecting sedimentation data over various time periods, and each reservoir is uniquely identified by a Reservoir ID (RID). The remaining three files consist of vector data—available in both shapefile and GeoJSON formats—representing the dam, reservoir, and catchment geometries for each reservoir.

### 3.1 Sedimentation Dataset

The Excel file GRILSS_data.xlsx contains sedimentation data for various reservoirs. The dataset includes 35 fields describing reservoir locations, characteristics, and the sedimentation that occurred over specific time periods. Each record also provides the original source URL, allowing users to explore or learn more about the reservoir or the sedimentation estimation method in detail. Missing values in the dataset are represented by blanks or empty spaces.

One of the fields includes comments that provide additional insights into each record or reservoir, such as its length and the sediment management techniques used. As detailed in Sect. 2.3, the method for estimating sediment volume is specified under the 'Survey Type' field and is categorized as 'BS' for bathymetric surveys, 'RS' for remote sensing, and 'NM' for numerical modeling methods. Fields that have been manually corrected, compared to the original source, are noted under the 'Fields Corrected' field. Table 1 outlines the details of each field in the dataset.

**Table 1: Overview and Description of Fields in the GRILSS Dataset (Minocha & Hossain, 2024)**

| S. No. | Field Name | Field Description | Units |
|---|---|---|---|
| 1. | GRILSS OID | A unique identifier for each sedimentation record in the dataset | - |
| 2. | GRILSS RID | A unique identifier for each reservoir in the dataset | - |
| 3. | Reservoir | The name of the reservoir | - |
| 4. | Country | The name of the country where the reservoir is located | - |
| 5. | HYBAS_ID | The HydroSHEDS HYBAS_ID for the major basin containing the reservoir | - |
| 6. | Major River Basin | The name of the major river basin, unique to each HYBAS_ID | - |
| 7. | Continent | The continent where the reservoir is situated | - |

| | | | |
|---|---|---|---|
| 8. | Capacity Loss Rate (%/year) | Annual rate of reservoir capacity loss, expressed as a percentage of the original built capacity. | %/year |
| 9. | Sedimentation Rate (MCM/year) | Annual rate of sediment deposition in the reservoir | Million cubic meters/year |
| 10. | Capacity Loss (%) | The percentage of the original built capacity (in MCM) that has been lost due to sediment accumulation | Percentage |
| 11. | Sedimentation Amount (MT) | The total amount of sediment deposited in the reservoir | Million tons |
| 12. | Sedimentation Amount (MCM) | The volume of sediment deposited in the reservoir | Million cubic meters |
| 13. | Sediment Bulk Density (ton/m³) | The estimated bulk density of the sediment deposited in the reservoir | Ton/m³ |
| 14. | Observed Duration (years) | The number of years over which the sedimentation amount was estimated | Years |
| 15. | Observation End Year | The year when the observed duration of sedimentation ended | - |
| 16. | Observation Start Year | The year when the observed duration of sedimentation began | - |
| 17. | Built Year | The year in which the reservoir was constructed | - |
| 18. | Original Built Capacity (MCM) | The reservoir's capacity at the time of construction | Million cubic meters |
| 19. | Catchment Area (km²) | The area of the reservoir's catchment | Square kilometers |
| 20. | Height (m) | The height of the reservoir's impoundment (dam) | Meters |
| 21. | Latitude | The latitude coordinate of the reservoir's dam | Degrees |
| 22. | Longitude | The longitude coordinate of the reservoir's dam | Degrees |
| 23. | Survey Type | The method used to estimate sedimentation volume in the reservoir [BS, RS, NM] | - |
| 24. | Type of Storage | The type of reservoir storage for which sedimentation volume is estimated [Gross or Live] | - |
| 25. | Comments | Additional information regarding the reservoir or the sedimentation estimation method | - |
| 26. | Original Source URL | The web URL to the original source from which the record was extracted | - |

| | | Names of fields that were corrected or added after manual inspection for the specific record | - |
|---|---|---|---|
| 27. | Fields Corrected | | |
| 28. | Fields Computed | Names of fields that were computed for the specific record | - |
| 29. | GDAT Dam Name | The name of the dam associated with the reservoir in the GDAT dataset | - |
| 30. | GDAT Feature_ID | The Feature_ID corresponding to the reservoir in the GDAT dataset | - |
| 31. | GRAND_ID | The GRAND_ID corresponding to the reservoir in the GRanD database | - |
| 32. | GRAND Wrong Location | A binary flag indicating if the dam coordinates are incorrect in the GRanD dataset | - |
| 33. | GDAT Wrong Location | A binary flag indicating if the dam coordinates are incorrect in the GDAT dataset | - |
| 34. | Dam Removed or Dried | A binary flag indicating if the reservoir has been removed or is dried | - |
| 35. | Creek Dam | A binary flag indicating if the dam is classified as a creek dam | - |

**3.2 Vector Dataset**

In addition to sedimentation data, the GRILSS dataset also includes associated vector data, as described in Sect. 2. This vector data comprises four files: 1. point locations of the dams associated with the reservoirs, 2. reservoir geometry polygons, 3. catchment geometry polygons, and 4. point locations representing snap points corresponding to dam positions for the creation of catchment boundary polygons. Each vector file contains attributes such as GRILSS RID, Reservoir name, and Country. Notably, the catchment vector file includes computed surface area for the reservoir

catchments, which may differ from the catchment area reported in the sedimentation data derived from the original source.

**4. GRILSS Dataset Overview**

The GRILSS dataset contains 1368 records for 1013 reservoirs worldwide. These reservoirs span 75 major river basins and 54 countries, representing a wide range of catchment types with varying physical characteristics such as slope,

climate, land use, land cover and soil type. Figure 1 highlights the 15 countries and major river basins with the highest number of reservoirs. The dataset also covers a wide range of dam construction periods, with few reservoirs dating back to the 1700s and 1800s. Most reservoirs, however, were constructed between 1950 and 2000, as shown in Fig. 2(b). The oldest reservoir in the dataset, Pareton in Spain, was built in 1713 (Erena et al., 2019) and has a capacity loss rate of 0.3% per year.

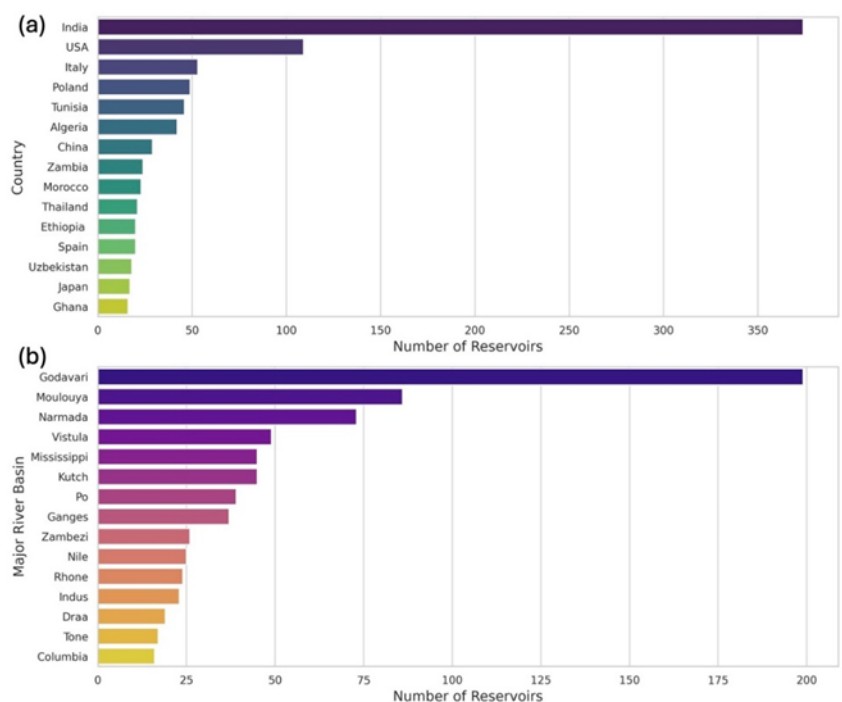


**Figure 1: The top fifteen (a) countries and (b) major river basins having the highest number of reservoirs in the GRILSS dataset.**

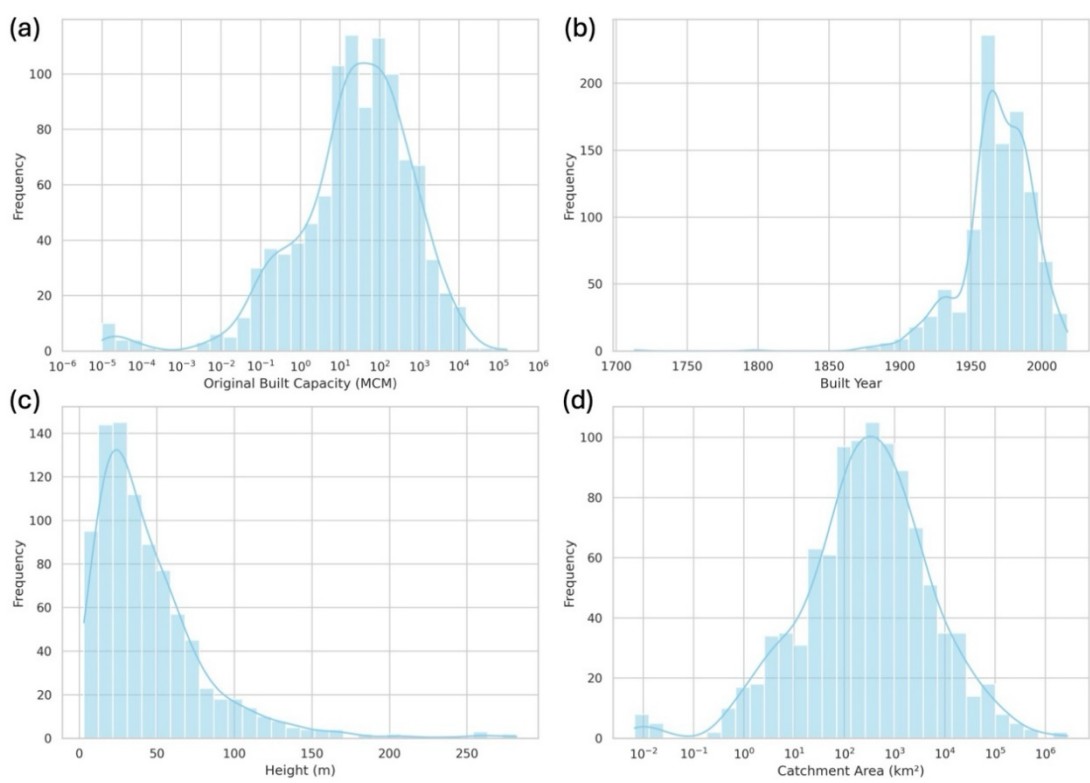

**Figure 2: Frequency distribution of reservoir attributes: (a) original built capacity, (b) construction year of dams, (c) dam height, and (d) catchment area. Each panel shows the distribution of these characteristics across the reservoirs in the dataset, with the blue line representing the kernel density estimate (KDE) of the distribution.**


In terms of survey methods, the dataset includes 1182 records based on bathymetric surveys (BS), 175 from remote sensing (RS), and 11 from numerical modeling (NM). The dataset also captures considerable variation in terms of the original built capacity of reservoirs, though reservoirs with capacities below 0.1 MCM are underrepresented (Fig. 2(a)). It includes 21 creek reservoirs, small dams constructed across creeks, all located in the United States. The

smallest reservoir, a creek sediment dam, has an original built capacity of just 9.9 m³. Additionally, 13 reservoirs in the dataset, all in the U.S., have either been removed or are part of dry dams. The largest reservoir in the GRILSS dataset is Lake Nasser in Egypt, with an original built capacity of 162 km³. Lake Nasser has one of the lowest capacity loss rates in the dataset, at $2.3 \times 10^{-4}$ % per year.

The height of the  dam is available for only 883 reservoirs in the dataset, collected from various sources as it was not consistently provided in the original data source. Figure 2(c) displays the frequency distribution of dam heights, with most dams being under 75 meters tall. The average height in the dataset is 45.58 meters. Cerro Prieto in Mexico, the tallest dam in the GRILSS dataset, stands at 282 meters with a capacity loss rate of 0.31% per year. In terms of catchment area, the dataset includes reservoirs with catchment sizes ranging from less than 1 km² to over 0.1 million

km². The distribution of catchment areas follows a normal pattern, with a mean of 14,759 km² (Fig. 2(d)).

**4.2 Capacity loss rate**

The capacity loss rate was first averaged across all records for each reservoir. From these reservoir-specific averages, the overall mean annual capacity loss rate for all reservoirs was determined to be 1.62% per year. Mayfair Lake in the USA had the highest observed loss rate at 102.24% per year. Some reservoirs even reported negative capacity loss

rates due to sediment management practices like dredging (Central Water Commission, 2020). Figure 3 shows the distribution of dams in GRILSS dataset and provides an overview of the rates at which reservoirs are losing capacity across different regions of the world. Figure 4(a) presents a histogram of the capacity loss rates, showing that the majority of reservoirs, 745 in total, have a loss rate of less than 1% per year.

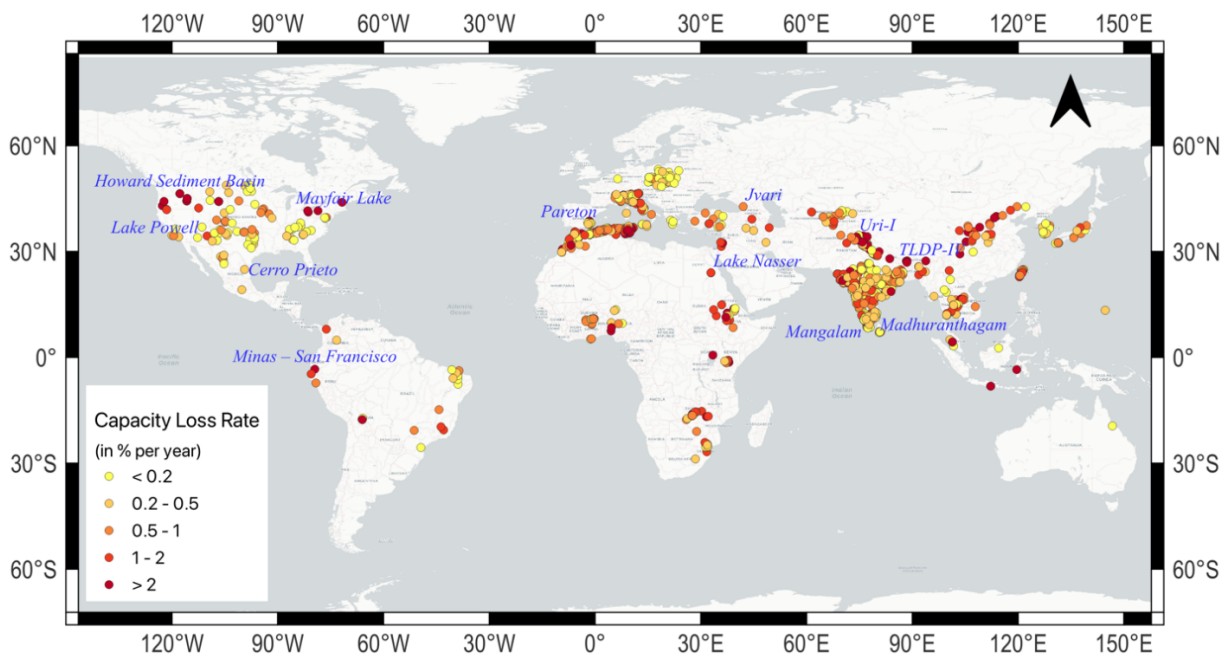


**Figure 3: Map displaying the locations of all reservoirs in the GRILSS dataset. Marker colors represent the capacity loss rate (% per year), with darker shades indicating higher loss rates. Names of select reservoirs referred to in the text or other figures are also labelled for additional information.**

### 4.2.1 Capacity loss rate by built capacity

Of the 1013 reservoirs, 174 (17.18%) have an initial built capacity of less than 1 million cubic meters (MCM), exhibiting an average capacity loss rate of 5.4% per year. This higher rate is largely attributed to creek dams, which can experience capacity loss rates as high as 132% per year due to their small size and short observed durations (around 1 year). The remaining 82.82% of reservoirs, with an initial capacity of 1 MCM or more, have an average capacity loss rate of 0.84% per year, consistent with global sedimentation rates reported by Basson (2009). Figure 4 (b) shows

the distribution of capacity loss rates for small (<1 MCM) and large reservoirs (≥1 MCM), while Table 2 summarizes these statistics for the GRILSS dataset.

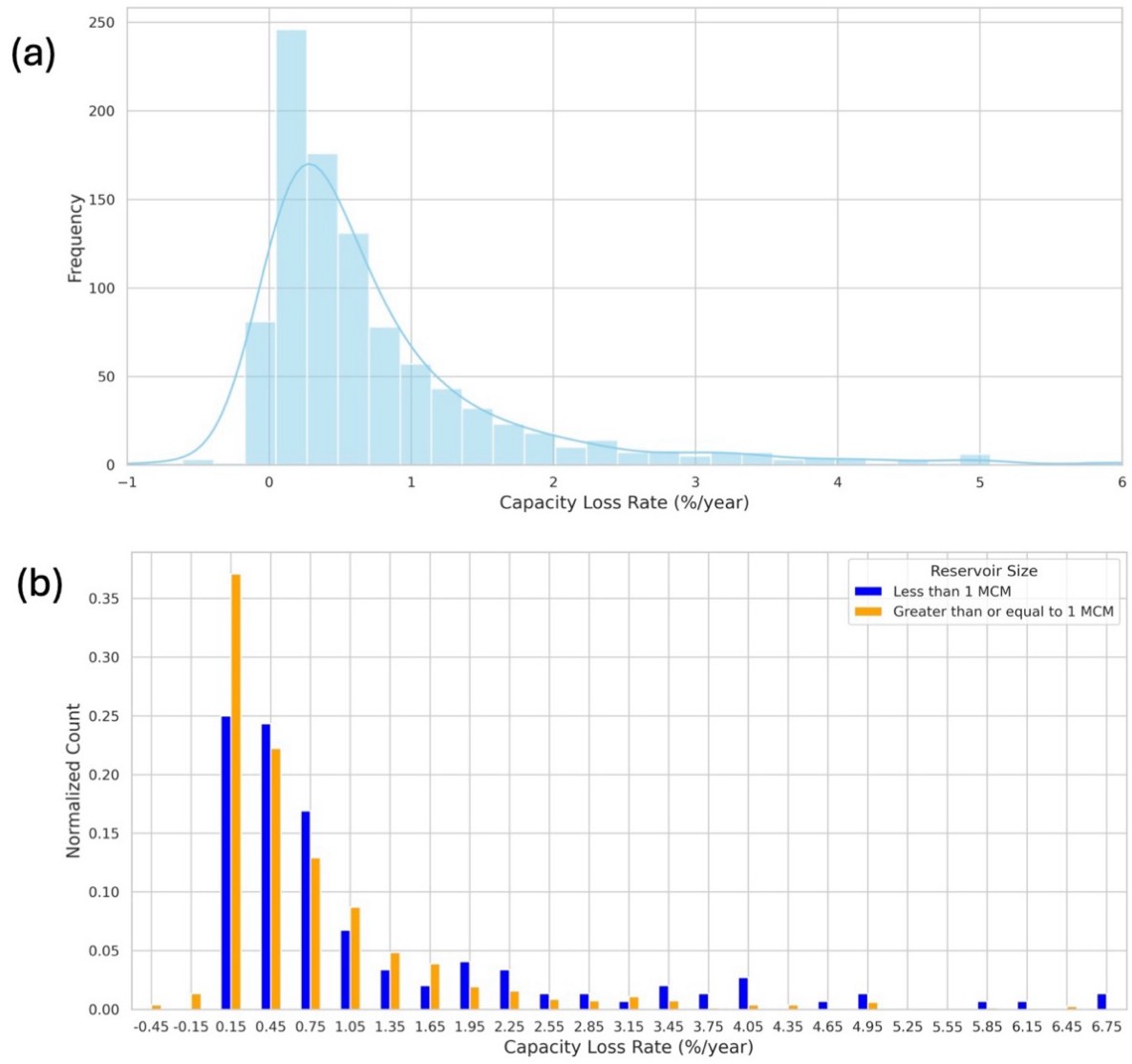

**Figure 4: (a) Frequency distribution of capacity loss rates for all reservoirs, with the blue line representing the kernel density estimate (KDE). (b) Distribution of capacity loss rates for reservoirs with an original built capacity of less than 1 MCM compared to those with a capacity of 1 MCM or more, normalized by the total count in each category.**


**Table 2: Statistics of annual capacity loss rate expressed as percentage of initial capacity of the reservoir (% a⁻¹)**

|  | Number of reservoirs | Min | Average | Median | Capacity-weighted mean | Max |
|---|---|---|---|---|---|---|
| Capacity <1 MCM | 174 | 0.0 | 5.4 | 0.76 | 1.2 | 102.24 |
| Capacity >1 MCM | 839 | -1.91 | 0.84 | 0.41 | 0.42 | 21.03 |

### 4.2.2 Capacity loss rate by observed duration

The observed duration for each record represents the time period over which sedimentation rates were estimated. Similar to the capacity loss rate, this duration was averaged across all records for a reservoir. As described in Sect 2.1, if only the capacity loss rate or sedimentation rate was provided without specific duration information, it was assumed to be one year.

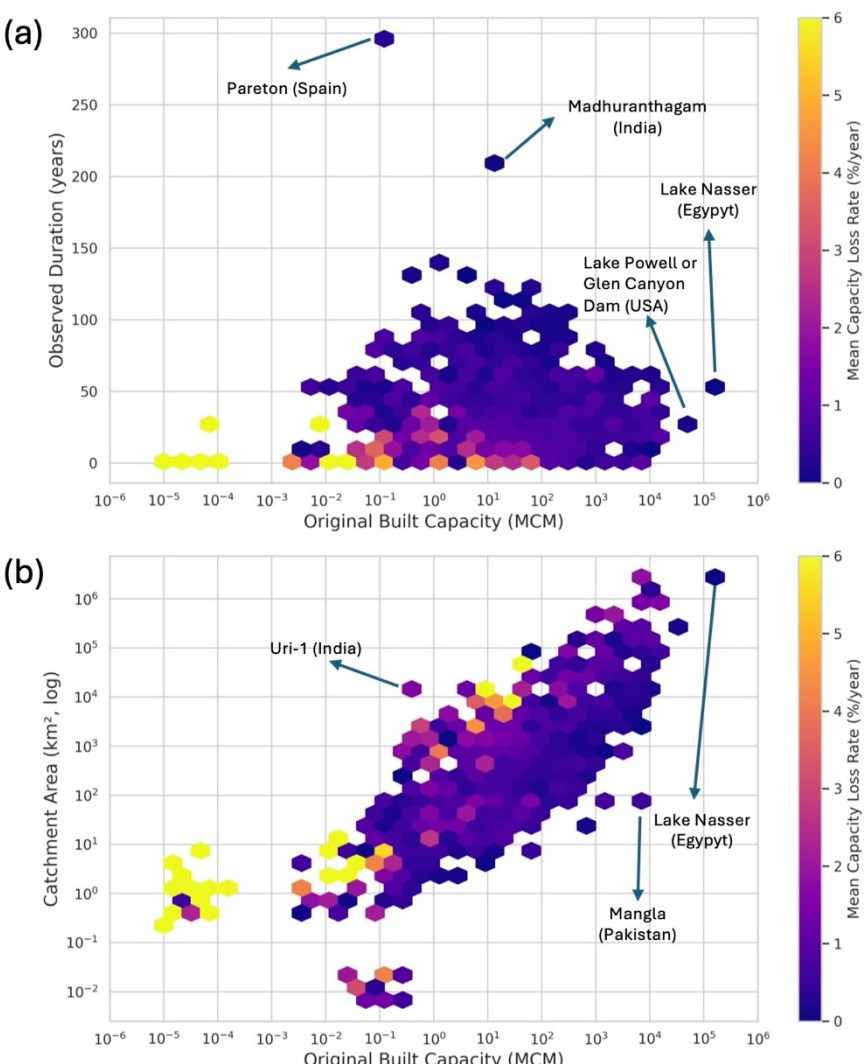

**Figure 5: Density scatter plot showing hexagonal bins corresponding to specific (a) built capacity and observed duration, and (b) original built capacity and catchment area of the reservoirs. The color gradient represents the mean capacity loss rate for all reservoirs within each hexagonal bin in both panels. Name of the reservoir, along with the country is mentioned for few outliers, where there was only one reservoir in the bin.**

Figure 5(a) illustrates that the capacity loss rate decreases as the observed duration increases. Additionally, for shorter
observed durations, smaller reservoirs tend to exhibit higher capacity loss rates, while larger reservoirs show lower rates. This relationship between observed duration and capacity loss rate may result from variable sedimentation rates

over time. Sedimentation rates are typically higher during significant storm events and floods; thus, a shorter observed duration is more likely to encompass a substantial portion of these events, leading to increased capacity loss rates.

### 4.2.3 Capacity loss rate by catchment area

Larger reservoirs tend to have larger catchment areas, while smaller reservoirs typically have smaller catchment areas, as illustrated in Figure 5(b). A similar relationship was observed by Walter et al. (2020) between catchment area and the reservoirs in terms of surface area. This figure also reinforces earlier observation that smaller reservoirs experience higher capacity loss rates. Notably, for a given built capacity, the capacity loss rate increases with the catchment area. This trend may be attributed to a greater availability of soil for erosion in large catchment areas.

### 4.2.4 Capacity loss rate by height

Figure 6 indicates that dam height does not significantly influence the capacity loss rate. However, it is observed that beyond a certain height—approximately 90 meters—the loss rate tends to remain low and does not reach high values. This pattern may again be attributed to larger capacity dams, which are generally taller and, consequently, exhibit lower loss rates.

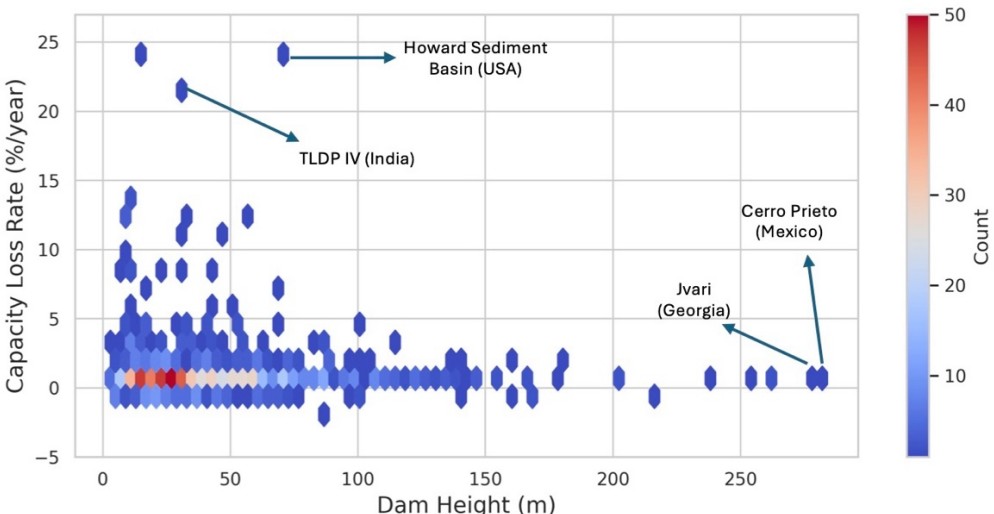


**Figure 6: Density scatter plot depicting the relationship between dam height (in meters) and capacity loss rate (% per year) within each hexagonal bin. The color gradient indicates the number of reservoirs contained in each bin. Name of the reservoir, along with the country is mentioned for few outliers, where there was only one reservoir in the bin.**

### 4.2.5 Capacity loss rate by built year

The GRILSS dataset includes two dams constructed in the 1700s, a few built in the 1800s, and many that extend into the twentieth and twenty-first centuries. This diverse range of construction years allows for an exploration of the relationship between the year a dam was built and its capacity loss rate. Figure 7 displays the median capacity loss rate for each year of construction, revealing an intriguing trend: reservoirs built more recently tend to have slightly higher capacity loss rates compared to older reservoirs. This subtle upward trend suggests that newly constructed
reservoirs may be losing capacity at an accelerated rate.

One possible explanation for this phenomenon is that sediment removal efforts are primarily focused on older reservoirs, while relatively newer ones have received less attention. Additionally, this long-term trend may be influenced by factors such as climate change and urbanization, and therefore needs to be further investigated.

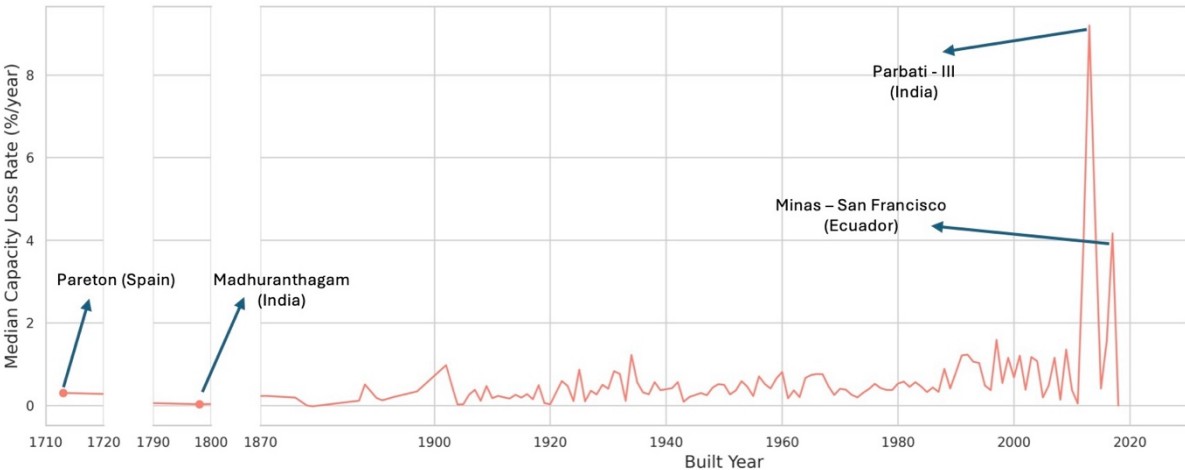

**Figure 7: Median capacity loss rate for each year of construction among the GRILSS reservoirs. The x-axis shows two distinct kinks corresponding to the two reservoirs built in the 18th century. Reservoir names are indicated for those constructed in that century and for any that exhibit anomalously high capacity loss rates relative to the overall trend.**

### 4.2.6 Capacity loss rate by Country

Total reservoir storage capacity is a critical concern for all countries as it determines the available freshwater for irrigation, domestic, and industrial use. To manage water resources effectively, it is essential for water managers to understand the average capacity loss rate within their country. Figure 8 presents the median capacity loss rates for 54 countries represented in the GRILSS dataset. Lighter shades, such as peach and light pink, indicate lower capacity loss rates (0–0.8% per year), while darker shades highlight countries experiencing higher losses (up to 7% per year). Countries where fewer than three reservoirs are available in the GRILSS dataset are outlined in red, marking the data for these areas as lower confidence. Notably, the United States and several countries in Southeast Asia show significantly higher capacity loss rates, indicating increased sedimentation challenges in these regions. This observation overall aligns with the findings of Perera et al. (2023), who also reported the highest capacity losses in the United States.

### 4.3 Sedimentation rate

While capacity loss rates are crucial for countries and water managers, understanding sedimentation rates is equally important, as they indicate which regions are experiencing more intense sedimentation. As discussed in Sect. 1, factors such as a region's topography and climate play a key role in influencing sedimentation. For each reservoir, the sedimentation rate was averaged across all records. Figure 9 presents the distribution of sedimentation rates across the GRILSS dataset. Notably, 17 reservoirs reported negative sedimentation rates, likely due to sediment removal efforts. The median sedimentation rate in the dataset is 0.14 MCM per year, while the mean rate is significantly higher at 2.79 MCM per year. This difference is due to a few reservoirs with exceptionally high sedimentation rates, such as

Xiaolangdi and Sanmenxia in the Yellow River basin, where rates reach around 200 MCM per year. In total, 60 reservoirs report sedimentation rates exceeding 10 MCM per year.

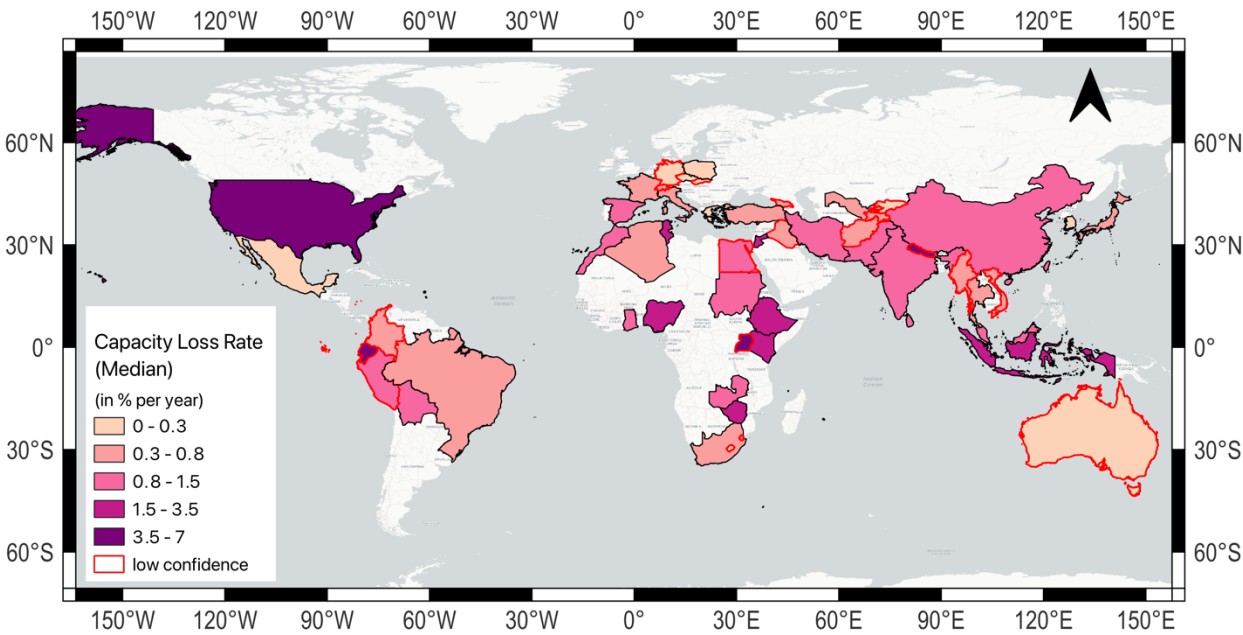

Figure 8: World map displaying the 54 countries covered by reservoirs in the GRILSS dataset. The color fill indicates the median capacity loss rate (% per year) for each country. Countries with fewer than three reservoirs are outlined in red, signifying lower confidence in the estimated loss rate for those regions.

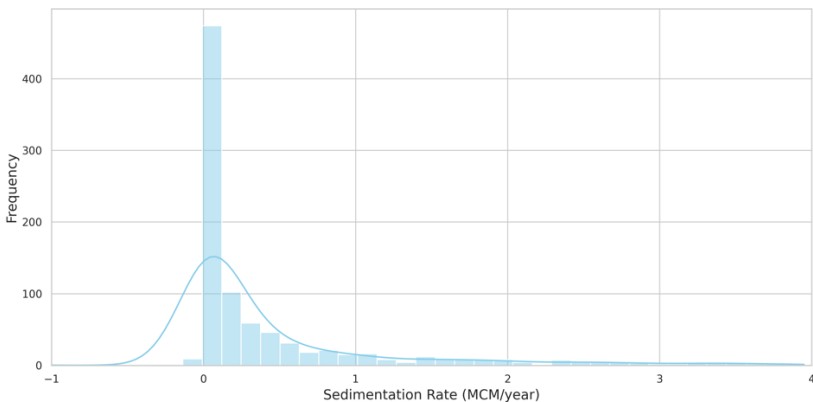

Figure 9: Frequency distribution plot of sedimentation rates (in MCM per year) for reservoirs in the GRILSS dataset, with a blue line representing the kernel density estimate.

### 4.3.1 Sedimentation rate by river basins

The 1013 reservoirs in the GRILSS dataset are distributed across 75 major river basins worldwide, each characterized by varying geography, climate, vegetation, population density, and topography. Figure 10 illustrates the median capacity loss rates for these river basins, with lighter shades of yellow representing lower sedimentation rates (0-4 MCM per year) and darker brown shades indicating higher rates (up to 56 MCM per year). The Parana River basin in South America exhibits the highest sedimentation rate at 56.2 MCM per year, followed by the Yellow River basin at 31.4 MCM per year. All river basins with sedimentation rates exceeding 8 MCM per year are labeled on the map.

River basins in North America and Europe exhibit lower sedimentation rates, whereas those in Asia and Africa tend to have higher rates.

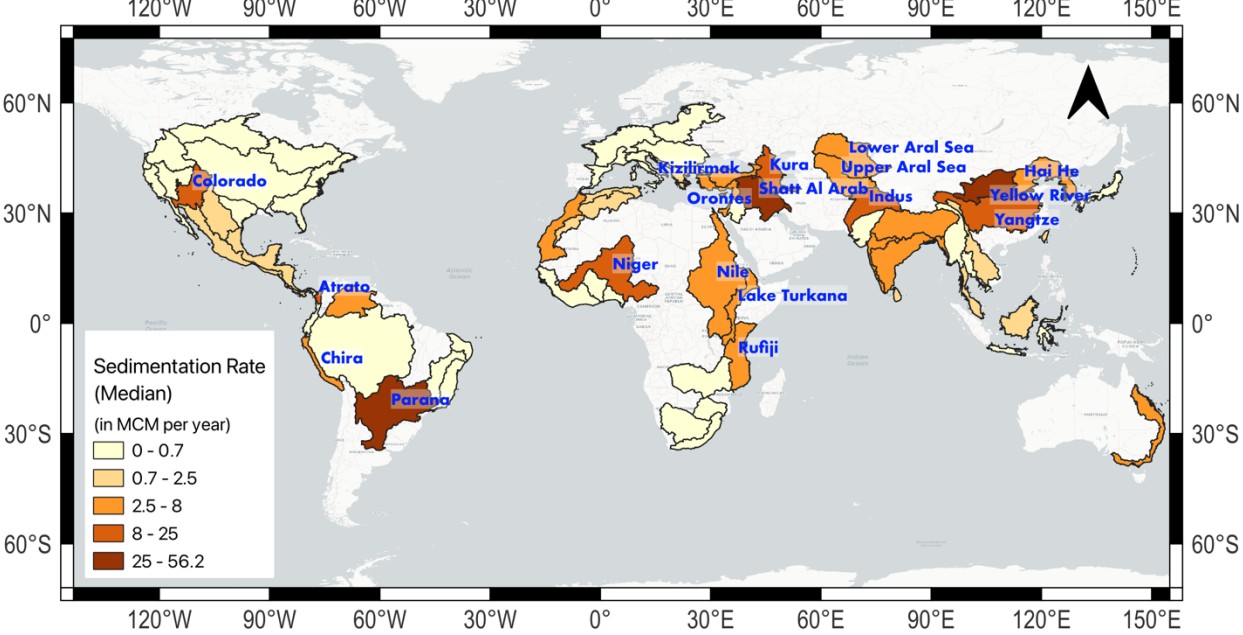

**Figure 10: Map depicting the median sedimentation rates of reservoirs across 75 major river basins represented in the GRILSS dataset. River basins with sedimentation rates of 8 MCM or more are labeled on the map.**

## 5. Discussion and Conclusion

Reservoirs play a critical role in meeting global freshwater demands for agriculture, domestic use, energy production, and industrial purposes by storing and regulating water supplies. Despite the continued construction of many new reservoirs worldwide, the total global storage capacity is in decline due to sedimentation. This loss of capacity is a growing concern, as sedimentation rates vary widely depending on factors such as the reservoir's location, catchment area, climate, topography, and soil characteristics. As a result, understanding the complexities of reservoir

sedimentation requires localized studies that account for these unique environmental conditions.

While numerous studies have estimated sedimentation rates for individual reservoirs or small regions, these efforts have largely remained fragmented, having disparate curation and confined to local scales. One notable global study by Wisser et al. (2013) attempted to estimate global storage capacity loss using observed reservoir sedimentation rates.

However, due to the lack of open-source data and the skewness of data toward reservoirs located in the United States (with over 1,000 records from the US and only 191 from the rest of the world), similar or follow up studies may have not been scalable. Thus, a lack of an organized, open-source dataset with a consistent curation practice that captures the reservoir sedimentation problem on a global scale, has limited the global use of sedimentation data.

Global dam datasets, such as GRanD (Lehner et al., 2011) and GDAT (Zhang & Gu, 2023), often include detailed reservoir attributes but generally lack sedimentation data. Similarly, data from the International Commission on Large Dams (ICOLD, 2019), while extensive, are not freely accessible in a readily usable tabular format. National-level efforts to compile reservoir sedimentation data, such as those for the United States (Gray et al., 2010), India (Central Water Commission, 2020), South Africa (Masadala et al., 2010), and Italy (Patro et al., 2022), have served as a

valuable resource. However, these datasets lack standardized curation protocols and are often inconsistent in the metrics they report. At the continental scale, Vanmaercke et al. (2014) compiled sediment yield data for 682 African catchments based on 84 publications and reports. While this work represents a significant contribution, it does not provide specific information on reservoir capacity loss.

To the best of our knowledge, GRILSS is the first comprehensive dataset to be able to track and compare sedimentation rates across reservoirs worldwide. GRILSS also enables the analysis of temporal variations in sedimentation rates driven by changes in hydrological conditions—an aspect not captured by earlier datasets. With 1013 reservoirs spanning 75 major river basins and 54 countries, GRILSS provides critical insights into the extent of the problem worldwide. Compiled from 143 diverse and traceable sources, including published articles, government documents,

and online databases, this dataset is a crucial resource for the global water community.

In an era where data and open access are essential, GRILSS opens new avenues for data-driven freshwater management on a global scale. Beyond sedimentation rates, the dataset also includes key reservoir attributes such as built capacity, dam height, and catchment area, along with information on the methods used to estimate sedimentation

and the time period of observations. Additionally, GRILSS provides vector data on reservoir geometries and catchments, making it a versatile tool for researchers and policymakers alike.

Initial analysis of the GRILSS dataset suggests a concerning trend of higher capacity loss rates in more recently constructed reservoirs, indicating that newer infrastructure may be more vulnerable to sedimentation and reservoir

operation efficiency. Additionally, we observed significant regional differences in sedimentation rates, with river basins such as the Parana and Yellow River experiencing some of the highest sedimentation rates globally. These insights underscore the need for targeted, region-specific sediment management strategies. It is important to note that this preliminary analysis primarily aims to highlight the strengths and potential of the GRILSS dataset, offering initial insights into sedimentation trends rather than providing a comprehensive critical appraisal of global sedimentation

patterns.

While the dataset provides valuable insights into reservoir sedimentation, certain limitations must be acknowledged. The underrepresentation of smaller reservoirs (<0.1 MCM) may bias global capacity loss rate estimates, as smaller reservoirs typically exhibit higher loss rates relative to their total storage capacity. Geographical coverage is also

uneven, partly due to reliance on English-language publications, which could overrepresent regions with better documentation. GDAT was used for georeferencing reservoirs, and dam coordinates were manually reviewed to

ensure accuracy. However, parameters like original built capacity and dam height, wherever missing, were also derived from GDAT. Since GDAT often lacks complete records for smaller dams, particularly in developing regions, it can introduce potential biases. These limitations emphasize the need for future efforts to include multilingual publications, validate data from multiple sources, and conduct more surveys for smaller reservoirs to achieve a more accurate representation of global reservoirs.


By organizing data on sedimentation and capacity loss that is also compatible with widely used existing reservoir datasets, GRILSS provides water managers, policymakers, and researchers a powerful dataset for planning and implementing better reservoir management practices. As the first dataset of its kind, GRILSS provides a foundation for future research into the myriad factors influencing reservoir sedimentation, including climate change, land use, and sediment management practices. Its global scope opens the gateway for collaborative data curation on reservoir sedimentation, enabling the community to bridge existing gaps to sustain a growing dataset for large-scale, global studies on reservoir sedimentation (Minocha & Hossain, 2024).



**Dataset Availability:** All data for GRILSS is hosted on the Open Science Forum at this link: https://osf.io/w4ug8/?view_only=6a595a2d635c48339c95502a5d1d6417   Minocha, S., & Hossain, F. (2024). GRILSS: Global Reservoir Inventory of Lost Storage by Sedimentation.

**Competing Interest:** The contact author has declared that none of the authors has any competing interests.

**Acknowledgements:** The work on the development of the GRILSS dataset was generously supported by the NASA Applied Science Program through grant 80NSSC22K0918 (Water Resources). The first author was supported by NASA Applied Science Program grant 80NSSC22K0918, National Science Foundation Graduate Traineeship program on Future Rivers and the Ivanhoe Foundation Fellowship.

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
