# Peer review of "GRILSS: Opening the Gateway to Global Reservoir Sedimentation Data Curation"

_Earth System Science Data, 2024_

## Author Response (AR1)

**POINT BY POINT RESPONSE TO REVIEWERS**

We appreciate the editor's help in providing us with timely reviews for our manuscript. We also appreciate time and effort the reviewer spent in helping us to improve the quality of our manuscript. We have considered each comment very seriously and performed additional and extensive analyses where appropriate to improve the quality of our manuscript. Below we summarize the key additional work we have undertaken to address the reviewers' concerns:

- We have added relevant citations throughout the manuscript to enhance readability and support our claims.
- We have explicitly acknowledged the uncertainties in the data, stemming from efforts to standardize sedimentation volumes and rates.
- We have highlighted the limitations of the dataset, particularly regarding potential biases in the representation of larger reservoirs and certain geographic regions.
- We have explicitly mentioned several global and regional datasets, highlighted their limitations and explained how GRILSS can fill the gaps.
- We have added a short discussion on the relative accuracy of the three methods for estimating sedimentation rates.
- Recommendations for future studies on reservoir sedimentation have been added, specifically advocating for the reporting of a minimum set of metrics for data curation.
- We have explicitly mentioned that the sedimentation volume in this study only considers sediment transported from outside the reservoir, resulting in capacity loss.
- We have clarified that the aim of the initial preliminary analysis is not to draw conclusions but rather to demonstrate the potential of the dataset.

In the section below, our response to each reviewer comment is shown in blue while the reviewer's comments are in black.

**SPECIFIC RESPONSES TO REVIEWER 1**

Q1. Some sentences in L25 lack references. For example, "However, these dams and their reservoirs pose significant environmental challenges, including the destruction of ecosystems, loss of biodiversity, disruption of aquatic life, greenhouse gas emissions, and barriers to fish migration." Please revisit and be specific, for instance, reservoirs pose significant environmental challenges (suggest 10.1007/s00382-024-07319-7), greenhouse gas emissions (suggest 10.3389/fenvs.2023.1304845), loss of biodiversity (10.3850/978-90-833476-1-5\_iahr40wc-p1339-cd), etc.

**Response:** Thank you for pointing that out. We have incorporated the suggested references and added additional citations to substantiate the points mentioned in L25. This should address the concerns and provide more specificity to the statement. The revised text on L25 is as follows:

"However, these dams and their reservoirs pose significant environmental challenges (Nguyen et al., 2024), including the destruction of ecosystems (Tundisi, 2018), loss of biodiversity (Wu et al., 2019), disruption of aquatic life (Morley, 2007), greenhouse gas emissions (Tran et al., 2023), and barriers to fish migration (Pelicice et al., 2015)."

Q2. Please be specific with this term "to implement sediment management techniques," e.g., of what?

Q3. L45 should be strengthened with more references. Please revisit and improve.

**Response:** Thank you for the comments. We have revised the text to include specific examples of sediment management techniques, such as sediment flushing, dredging, and watershed management. This addition clarifies the term and makes it more precise and informative for readers. Additionally, we have incorporated more references to strengthen the statement. The revised text on L45 is as follows:

"One of the primary solutions is to implement sediment management techniques, such as sediment flushing, dredging, and watershed management, which requires a thorough understanding of sedimentation rates in reservoirs worldwide (Morris, 2014; Kondolf et al., 2014; George et al., 2017)."

Q4. Please provide reference(s) for this claim "While indirect methods are less expensive, they rely on the availability of in-situ data, such as soil moisture and soil type for numerical models."

**Response:** Thank you for the comment. We have added references supporting the claim. The additional citations further substantiate the statement. The revised line is as follows: "While indirect methods are less expensive, they rely on the availability of in-situ data, such as soil moisture and soil type for numerical models, or reservoir water elevation data, often obtained through in-situ or altimetry-based measurements, for remote sensing applications (Ghosh et al., 2024; de Oliveira Fagundes et al., 2020; Dutta, 2016).

Q5. The authors acknowledge the highly variable nature of the source data, with sedimentation reported in different units (MCM, MT, percentage loss) and over different time scales. While they attempt to standardize this with conversions and assumptions (e.g., assuming a 1-year period when duration is unspecified, using estimated bulk density), these introduce significant uncertainties. The accuracy of calculated sedimentation volumes hinges heavily on the availability and reliability of bulk sediment density data, which the manuscript admits is often lacking. This inherent inconsistency in the underlying data makes comparisons between reservoirs challenging and potentially misleading. The assumption of a one-year sedimentation period when not specified is arbitrary and likely inaccurate in many cases, further contributing to the uncertainty.

**Response:** Thank you for your insightful comment. We agree that the inherent variability and inconsistency in the source data present challenges and introduce uncertainties in sedimentation estimates. To address this, we have clarified the potential implications of these uncertainties and highlighted them explicitly in the manuscript. Specifically, we have revised the text to:

1. Acknowledge the uncertainties: We emphasize the variability in sedimentation reporting units (MCM, MT, percentage loss) and the assumptions made for standardization, such as bulk density estimations and the assumption of a one-year sedimentation period when duration is unspecified.

- 2. **Contextualize the assumptions:** While we acknowledge that the assumption of a oneyear sedimentation period is arbitrary and may not reflect the actual duration, we clarify that this assumption was used only to calculate sedimentation volume from sedimentation rate, and not vice versa. Therefore, sedimentation rates remain unaffected by this assumption and can be reliably used for comparing between reservoirs.
- 3. **Highlight future improvements:** We suggest the importance of future efforts to improve consistency in reporting data, including bulk sediment density data and more detailed sedimentation time scales.

We hope this revision provides greater clarity and transparency regarding the limitations of the data and the assumptions employed. Thank you again for raising this important point.

The following paragraph has been added in the end of section 2.1:

"While efforts were made to standardize sedimentation data through conversions and assumptions, such as assuming a one-year sedimentation period when not specified or estimating bulk sediment density, these approaches inevitably introduce uncertainties. It is important to note that the assumption of a one-year sedimentation period affects essentially sedimentation volume calculations and not sedimentation rates. Thus, sedimentation rates should be used for inter-reservoir comparisons. The reliance on such assumptions reflects the limitations of the available data and underscores the need for more robust and consistent reporting practices globally. Consequently, trends and comparisons derived from these data should be interpreted with caution."

Q6. While the dataset includes a large number of reservoirs, the manuscript acknowledges underrepresentation of smaller reservoirs (<0.1 MCM). This biases the overall analysis of capacity loss rates, as smaller reservoirs tend to have higher rates. The geographical distribution of data also appears uneven. While the goal is global coverage, the methodology hints at potential over-representation of data from regions where more readily accessible English-language publications exist. This bias could skew global assessments of sedimentation patterns and influence conclusions about regional variations. The heavy reliance on GDAT for georeferencing also introduces potential biases if GDAT itself has systematic errors or uneven global representation in which I found the GDAT often lacks comprehensive dam records, particularly for small dams and reservoirs and many regions, especially in developing countries, have poor or outdated records, leading to incomplete datasets.

**Response:** Thank you for the detailed and constructive comment. We acknowledge the concerns raised regarding the dataset's representation and potential biases. Below, we outline how we have addressed these points in the manuscript:

**1. Underrepresentation of smaller reservoirs (<0.1 MCM):**

We acknowledge the underrepresentation of smaller reservoirs in the dataset and its potential to bias the global capacity loss rate analysis, as smaller reservoirs tend to exhibit higher loss rates relative to their total storage capacity. We have highlighted this limitation in the manuscript and discussed how it might influence the findings. Future efforts to expand the dataset and include smaller reservoirs are essential for a more balanced global assessment.

**2. Geographical distribution and potential language bias:**

We recognize the geographical bias stemming from reliance on English-language publications. This limitation has been explicitly noted in the manuscript, along with suggestions for addressing it in the future, such as incorporating more multilingual datasets.

**3. Reliance on GDAT for georeferencing and other parameters:**

While we relied on GDAT for georeferencing, it is important to note that the sedimentation data was collected independently of GDAT. Additionally, we manually reviewed and cross-checked dam coordinates obtained from GDAT for accuracy and corrected errors to the best of our knowledge. However, it is important to note that other parameters such as original built capacity and dam height were also taken from GDAT wherever it was not available in the original source. Since GDAT often lacks complete records for smaller dams and reservoirs, particularly in developing regions, this could introduce potential biases in the dataset. We have revised the text to explicitly address these points.

The following paragraph has been added in the section 5:

"While the dataset provides valuable insights into reservoir sedimentation, certain limitations must be acknowledged. The underrepresentation of smaller reservoirs (<0.1 MCM) may bias global capacity loss rate estimates, as smaller reservoirs typically exhibit higher loss rates relative to their total storage capacity. Geographical coverage is also uneven, partly due to reliance on English-language publications, which could overrepresent regions with better documentation. GDAT was used for georeferencing reservoirs, and dam coordinates were manually reviewed to ensure accuracy. However, parameters like original built capacity and dam height, wherever missing, were also derived from GDAT. Since GDAT often lacks complete records for smaller dams, particularly in developing regions, it can introduce potential biases. These limitations emphasize the need for future efforts to include multilingual publications, validate data from multiple sources, and conduct more surveys for smaller reservoirs to achieve a more accurate representation of global reservoirs."

Q7. The manuscript identifies several factors affecting sedimentation, including topography, climate, land use, and reservoir shape. However, the analysis primarily focuses on simple bivariate relationships (e.g., capacity loss vs. catchment area, capacity loss vs. dam height). This neglects the complex interplay of these factors and their potential non-linear interactions. For example, the relationship between catchment area and sedimentation is likely mediated by land use, soil type, and precipitation patterns. Without accounting for these complexities, the analysis lacks the depth needed to draw robust conclusions about the drivers of sedimentation. The use of readily available digital elevation models (DEMs) for catchment delineation, especially at coarser resolutions (90m), might not accurately capture the true catchment boundaries, particularly in complex terrains or flat areas, influencing the calculation of catchment areas and related analyses.

**Response:** Thank you for your valuable comment. We acknowledge the complexity of the factors influencing sedimentation, including their potential non-linear interactions. However, the primary goal of this manuscript is to introduce the GRILSS dataset. An initial overview of the data through simple univariate and bivariate analyses has been provided to give the readers an idea of how the reservoirs in the GRILSS dataset are distributed in terms of original built capacity, height, catchment area, and capacity loss rate. It also helps in understanding the reliability of the data, while offering a broad perspective on global sedimentation patterns and capacity loss rates. Another purpose of these limited analysis is to show the readers the potential of the GRILSS dataset and the myriad of studies that one could pursue with it. By no means, these analyses are exhaustive as the goal of the paper is to introduce the dataset and not the research questions that can be answered with it. A more in-depth analysis of the complex interplay between the various factors like topography, climate, land use, and reservoir shape, is beyond the scope of this work but could be explored in future studies.

Regarding catchment delineation, the provided catchment shapefiles are intended as supplementary information for users. We recognize that in cases where high-resolution DEMs are required, users are encouraged to use their own catchment geometries based on the specific needs of their work, and we have updated the manuscript to reflect this. The following text has been added in the end of section 2.5:

"The catchment shapefiles are provided as supplementary information for users. However, in cases where high-resolution DEMs are required, users are encouraged to use their own catchment geometries to suit the specific needs of their work."

**SPECIFIC RESPONSES TO REVIEWER 2**

While this is a potentially interesting paper, and relevant to the journal to which it was submitted, there are a number of limitations and omissions that, in my opinion, need to be addressed before acceptance. First, there is no recognition of other global databases including, for example, that available from the International Commission on Large Dams (ICOLD) and that published by Vanmaerck in 2014 for example (Earth-Science Reviews 136 (2014) 350–368). Regionally specific datasets like those for South Africa by Rooseboom and Msadala are also not cited. While the study by Vanmaercke focuses on sediment yields it includes a comprehensive assessment of the quality of the data contained in the database. As the current study includes a wide range of timescales it fails to acknowledge the limitations posed by this issue in relation to temporal variability caused by changes in hydrological conditions. Some mention of other databases is needed to demonstrate what their limitations are and what added value is provided by the current database.

**Response:** Thank you for bringing this point to our attention. While we briefly discussed regional reservoir sedimentation datasets from the United States and India in the Introduction, we acknowledge that we did not include other global datasets such as the one available from ICOLD. This omission was primarily due to ICOLD's limited reservoir sedimentation data and restricted accessibility. However, we recognize the importance of citing these datasets to

contextualize our work better and to highlight the unique contributions and strengths of our dataset.

We also appreciate your observation regarding our dataset's potential to track temporal variability in sedimentation rates due to changing hydrological conditions. Including this aspect further underscores the utility and limitations of GRILSS. To address these valuable suggestions, we have added references to relevant global and regional datasets in the Discussion section. We have also discussed their limitations and how GRILSS bridges some of these gaps, thus advancing global sedimentation research. The following text has been incorporated into the section 5:

"Global dam datasets, such as GRanD (Lehner et al., 2011) and GDAT (Zhang & Gu, 2023), often include detailed reservoir attributes but generally lack sedimentation data. Similarly, data from the International Commission on Large Dams (ICOLD, 2019), while extensive, are not freely accessible in a readily usable tabular format. National-level efforts to compile reservoir sedimentation data, such as those for the United States (Gray et al., 2010), India (Central Water Commission, 2020), South Africa (Masadala et al., 2010), and Italy (Patro et al., 2022), have served as a valuable resource. However, these datasets lack standardized curation protocols and are often inconsistent in the metrics they report. At the continental scale, Vanmaercke et al. (2014) compiled sediment yield data for 682 African catchments based on 84 publications and reports. While this work represents a significant contribution, it does not provide specific information on reservoir capacity loss.

To the best of our knowledge, GRILSS is the first comprehensive dataset to be able to track and compare sedimentation rates across reservoirs worldwide. GRILSS also enables the analysis of temporal variations in sedimentation rates driven by changes in hydrological conditions—an aspect not captured by earlier datasets."

I am also very concerned about the use of uncalibrated models like USLE to fill in missing data. There is no acknowledgement of the issues associated with a model developed for hillslope plots applied to very large catchments. There are known to be huge discrepancies between modelled and measured estimates but this is simply ignored in the paper and again needs to be addressed. Essentially, the authors need to recognise the limitations of what they have achieved by compiling this data set. I believe this would strengthen the paper by giving the readers an honest and detailed account of the limitations. It also means that end users are aware of potential limitations before they use it.

**Response**: Thank you for bringing this concern to our attention. We sincerely appreciate the opportunity to clarify that no models, such as USLE, were applied to fill in missing data in our study. The modeled observations of sedimentation rates included in GRILSS were directly sourced from published literature and represent only a very small portion of the dataset (10 out of 1,368 observations). Nevertheless, we agree with your valuable suggestion that the limitations of GRILSS should be clearly acknowledged, especially those arising from uncertainties and biases. Doing so will ensure that users have a clear and transparent understanding of the dataset's scope and its possible limitations.

The following paragraph has been added in the section 5:

"While the dataset provides valuable insights into reservoir sedimentation, certain limitations must be acknowledged. The underrepresentation of smaller reservoirs (<0.1 MCM) may bias global capacity loss rate estimates, as smaller reservoirs typically exhibit higher loss rates relative to their total storage capacity. Geographical coverage is also uneven, partly due to reliance on English-language publications, which could overrepresent regions with better documentation. GDAT was used for georeferencing reservoirs, and dam coordinates were manually reviewed to ensure accuracy. However, parameters like original built capacity and dam height, wherever missing, were also derived from GDAT. Since GDAT often lacks complete records for smaller dams, particularly in developing regions, it can introduce potential biases. These limitations emphasize the need for future efforts to include multilingual publications, validate data from multiple sources, and conduct more surveys for smaller reservoirs to achieve a more accurate representation of global reservoirs."

Line 89 Was this review done as a systematic review using specified search engines and search terms? This is becoming a standardised way of ensuring published literature is not missed but the grey literature is of course more complex and often not in the public domain. Exactly how were the reviewed papers sorted into those that were or were not relevant. Did you have a set of criteria to do this? If so, what were they?

**Response:** Thank you for raising the question regarding the systematic nature of the literature review. We acknowledge that the review process was not entirely systematic. For this study, search engines such as Google, Google Scholar, and ResearchGate were utilized to identify relevant literature, using primary search terms such as "reservoir capacity loss," "reservoir sedimentation," "bathymetric surveys," and "sedimentation rate," among others. The classification of relevant literature was based on whether the document contained any specific data on reservoir sedimentation metrics for any reservoir.

As you rightly noted, conducting a systematic review can be particularly challenging when dealing with grey literature, which is often not publicly accessible or indexed in conventional databases. This limitation required adopting a more flexible approach to ensure the inclusion of valuable but less readily available data sources. While this method lacks the rigor of a fully systematic review, it was considered suitable for the scope of this study, given its focus on extracting sedimentation data from diverse and often unconventional sources. Grey literature played a crucial role in increasing the comprehensiveness of the dataset, contributing unique data points that may not be available in indexed publications.

To address this issue, we have revised relevant text as follows in section 2.1:

"The GRILSS dataset (Minocha & Hossain, 2024) was compiled through an extensive literature review of published articles, theses, government documents, and websites, some of which were in languages other than English (Estrada et al., 2015). The literature review was not completely systematic. Search engines such as Google, Google Scholar, and ResearchGate were utilized with keywords like "reservoir capacity loss," "reservoir sedimentation," "bathymetric surveys," and "sedimentation rate," among others. Conducting a fully systematic review proved challenging due to the inclusion of grey literature, which is often not indexed in standard databases or publicly accessible. After carefully reviewing hundreds of papers and documents, the dataset was created by manually compiling data from 143 different sources using Microsoft Excel. Only sources containing specific sedimentation data related to reservoirs were included. This thorough process ensured that the dataset covers sedimentation information from a wide range of regions."

**Line 92 containing not contained**

**Response:** Thank you for pointing that out. We have made the necessary correction.**

Line 120 It would be useful based on your experience to ask when reservoir data are reported that a minimum number of metrics should be reported for each reservoir as a minimum requirement for acceptance. There is a lot we can learn from your review but it needs the authors to think this through and make recommendations for future studies if we are to improve future datasets.

**Response:** Thank you for your excellent suggestion. We agree that setting minimum reporting standards for reservoir data would greatly improve the consistency and quality of future datasets. Based on our experience, we suggest that sedimentation volume should always be reported in MCM (million cubic meters), along with the specific time frame during which the sedimentation occurred. Additionally, we recommend explicitly reporting dam attributes such as the original built capacity, built year, and dam coordinates. While the spelling of reservoir names may vary across regions or sources, the dam coordinates remain unique, which can help to ensure the correct identification and comparison of reservoirs.

We have incorporated these recommendations into the manuscript in the end of section 2.1 as follows:

"For consistency and accuracy in reporting reservoir sedimentation data, we recommend the inclusion of minimum key metrics for each reservoir. These should include the sedimentation volume in million cubic meters (MCM) and the time frame over which sedimentation occurred. Additionally, it is important to indicate whether the reservoir employs any sediment management techniques. Key dam attributes such as the original built capacity, construction year, and precise dam coordinates should also be reported. Although variations in the spelling of reservoir names may arise across different sources, the inclusion of coordinates offers a unique and reliable method for identifying and comparing reservoirs, thus improving data consistency and reliability in future studies."

Lines 160-175 It would be useful if authors could comment on the relative accuracy of these three methods. In my experience different models give wildly different estimates of sediment transport so reservoir storage estimates come with potentially very large errors. This has implications from any analysis of the data.

**Response:** Thank you for your valuable suggestion. We agree that providing insight into the relative accuracy of the three methods would be useful for readers. You have rightly pointed that the accuracy of these methods can vary significantly depending on several factors. For numerical modeling, the accuracy is largely determined by the quality of the input data and the assumptions

made within the model, with more complex models generally offering better accuracy. For satellite remote sensing, accuracy is influenced by factors such as the resolution of the satellite imagery, the method used to estimate the surface area of the reservoir, and whether the elevation data is in situ or derived from altimetry. Despite these varying factors, it is generally observed that bathymetric surveys tend to offer the highest accuracy, followed by remote sensing, and then numerical modeling.

We have added a paragraph on this in the manuscript in section 2.3 to provide clarity on these aspects:

"In terms of relative accuracy, the methods for estimating sedimentation rates and reservoir storage vary depending on the approach and data used. Bathymetric surveys are generally considered to provide the most accurate estimates, as they directly measure the reservoir's depth and volume. Satellite remote sensing accuracy is influenced by factors such as image resolution, the method used to estimate the surface area of the reservoir, and whether elevation data is in situ or derived from altimetry. Numerical modeling, on the other hand, depends heavily on the accuracy of input data and the assumptions inherent in the model. While these methods have inherent limitations, bathymetric surveys typically yield the highest accuracy, followed by remote sensing and numerical modeling, respectively (Nyikadzino & Gwate, 2021; Gao, 2009)."

Figure 10 provides global estimates of reservoir sedimentation. I find some of these estimates extremely surprising especially the slow rates of sedimentation in Southern Africa which is recognised as a severly degraded landscape and where modelled and measured estimates of sediment yield (derived from reservoir sedimentation rates in manhy cases) are amongst the worlds highest. Again, this suggests to me that the data base has not been critically appraised by the authors in the light of significant amounts of published data. The analysis of the data contained in this database as presented in this paper does not stand up to detailed scrutiny in the light of other published data.

**Response:** Thank you for your insightful comment. We truly appreciate your concern regarding the sedimentation rates in Southern Africa. We would like to clarify that our role in this study was to compile data from various sources without altering or adjusting it in any way. To maintain full transparency, we have ensured that the source URL for each observation is included in the dataset.

Regarding South Africa, the capacity loss rates presented in Figure 3 appear consistent with the annual capacity loss rate distribution reported in Table 1.1.2 of Msadala et al. (2010). Additionally, it is important to note that Figure 10 shows sedimentation rates in Million Cubic Meters (MCM) per year. In reservoirs with small catchment areas, sediment yield can be much higher, which could explain some of the discrepancies observed. Vanmaercke et al. (2014) also noted that many sediment yield estimates derived from reservoir sedimentation rates in Southern Africa are for smaller catchments, which may help explain some of the differences you've highlighted.

Finally, while we agree that a critical appraisal of the data is essential, it is worth noting that the aim of this preliminary analysis was not to draw strong conclusions about global sedimentation

patterns but rather to provide an overview of the potential of the GRILSS dataset and to highlight the available data.

We have revised relevant text in the section 5 of the manuscript. We hope this explanation clarifies the intention behind our presentation.

"Initial analysis of the GRILSS dataset suggests a concerning trend of higher capacity loss rates in more recently constructed reservoirs, indicating that newer infrastructure may be more vulnerable to sedimentation and reservoir operation efficiency. Additionally, we observed significant regional differences in sedimentation rates, with river basins such as the Parana and Yellow River experiencing some of the highest sedimentation rates globally. These insights underscore the need for targeted, region-specific sediment management strategies. It is important to note that this preliminary analysis primarily aims to highlight the strengths and potential of the GRILSS dataset, offering initial insights into sedimentation trends rather than providing a comprehensive critical appraisal of global sedimentation patterns."

**SPECIFIC RESPONSES TO REVIEWER 3**

1. Due to the destruction of the reservoir banks, sediments enter the reservoir, filling it, but at the same time, the capacity of the reservoir increases due to the retreat of the bank. Accordingly, this should be taken into account in formula 3, since there are a number of reservoirs where there is a significant retreat of the banks.

**Response:** Thank you for pointing that out. It is a great point, and we truly appreciate your observation. When sedimentation volume is reported in the case of reservoirs, it is usually estimated by comparing the bathymetry of reservoirs at two different times. Therefore, the volume of storage lost in the reservoir is said to be due to sedimentation. This sedimentation primarily occurs from the transport of sediments from the catchment to the reservoir and their subsequent settling at the bottom. Consequently, the sedimentation volume is considered the same as the capacity lost in such cases. However, we agree that this point should be explicitly clarified in the manuscript.

Thus, we have added the following text in section 2.1 to address this aspect:

"In the context of sedimentation volume, it should be noted that this is typically derived by comparing reservoir bathymetry data collected at two different times. The capacity lost is attributed to the sedimentation caused by sediment transport from the catchment and deposition within the reservoir. While this approach assumes sedimentation volume is primarily from catchment-derived input, it does not account for sedimentation resulting from the transport of sediments within the reservoir itself, including processes like the retreat of reservoir banks. Future studies addressing sedimentation dynamics should explicitly consider such cases when relevant." 2. Figure 8 seems unnecessary, since the reliability of estimates for many countries is insufficient, and most importantly, the data is highly uneven across countries.

**Response**: Thank you for your feedback. We recognize that data for many countries is unavailable, and, in some cases, the reliability is low. However, Figure 8 was included as the most effective way to present a geographical map of countries with capacity loss rates, highlighting the 54 countries for which reservoir sedimentation data exists in GRILSS. That said, we acknowledge your concern and agree that it is not a strong figure from an analytical perspective.

To address this, we have added the following text to Section 5:

" It is important to note that this preliminary analysis primarily aims to highlight the strengths and potential of the GRILSS dataset, offering initial insights into sedimentation trends rather than providing a comprehensive critical appraisal of global sedimentation patterns."

3. Reservoirs that were periodically sediment-dredging should be allocated to a separate group. Data on sedimentation of such reservoirs distorts the general patterns of influence of various factors.

**Response:** Thank you for your valuable comment. We agree that reservoirs subjected to periodic sediment-dredging may exhibit different sedimentation dynamics, which could distort the general patterns observed in the data. While we have not specifically categorized dredged reservoirs in this study, we have noted this information in the comments where available. We recognize the importance of distinguishing dredged and non-dredged reservoirs in future analyses and will ensure this is considered in subsequent work.

To address this, we have added the following text in Section 2.1:

[revised manuscript text omitted]